# Differences in the Sublethal Effects of Sulfoxaflor and Acetamiprid on the *Aphis gossypii* Glover (Homoptera: Aphididae) Are Related to Its Basic Sensitivity Level

**DOI:** 10.3390/insects13060498

**Published:** 2022-05-26

**Authors:** Wei Wang, Qiushi Huang, Xiaoxia Liu, Gemei Liang

**Affiliations:** 1State Key Laboratory for Biology of Plant Diseases and Insect Pests, Institute of Plant Protection, Chinese Academy of Agricultural Sciences, Beijing 100193, China; wlzforever2004@sina.com (W.W.); hqs18514344231@163.com (Q.H.); 2Department of Entomology, China Agricultural University, Beijing 100193, China; liuxiaoxia611@cau.edu.cn; 3Key Laboratory of Integrated Pest Management on Crops in Northwestern Oasis, Ministry of Agriculture and Rural Affairs, Institute of Plant Protection, Xinjiang Academy of Agricultural Sciences, Urumqi 830091, China

**Keywords:** sublethal effects, *Aphis gossypii*, sulfoxaflor, acetamiprid, life table

## Abstract

**Simple Summary:**

The sublethal effects of insecticides are not only environmentally risky to arthropods but may also promote resistance evolution. Sublethal effects are influenced by factors such as the type of insecticide, sublethal concentration, and type of pest. This study evaluated the sublethal effects of sulfoxaflor and acetamiprid on two field cotton aphid (*Aphis gossypii*) populations with different genetic backgrounds. For acetamiprid, a significant negative sublethal effect of an LC_25_ concentration of acetamiprid on longevity and fecundity was observed in the F_0_ generation of Jinghe, and a significant negative sublethal effect occurred in the F_1_ and F_2_ generations of Yarkant, some biological traits of which were significantly degraded. However, in terms of biological traits, significant stimulative sublethal effects of an LC_25_ concentration of sulfoxaflor were observed in the F_0_ generation of Jinghe and the F_1_ generation of Yarkant. These experimental results demonstrate that sulfoxaflor and acetamiprid have different sublethal effects on *A. gossypii* that vary depending on the generation. Moreover, the sublethal effects of an insecticide may be influenced by the genetic background and resistance levels of *A. gossypii*. Our findings are useful for assessing the overall effects of sulfoxaflor and acetamiprid on *A. gossypii*.

**Abstract:**

The cotton aphid, *Aphis gossypii*, is an important insect pest of many crops around the world, and it has developed resistance to a large number of frequently used insecticides. The sublethal effects of insecticides not only have an environmental risk to arthropods but also have the potential to promote resistance evolution. The sublethal effects (inhibitory or stimulatory) are influenced by many factors, such as the type of insecticide, sublethal concentrations, pest species, and others. In this study, the sublethal effects of sulfoxaflor and acetamiprid on *A. gossypii* were compared using two field-collected populations. The results show that sulfoxaflor was more toxic than acetamiprid against *A. gossypii* in both populations, the LC_50_ concentrations of acetamiprid and sulfoxaflor were 6.35 and 3.26 times higher, respectively, for the Jinghe population than for Yarkant. The LC_25_ concentration of acetamiprid significantly reduced adult longevity and fecundity in exposed adults (F_0_) of the Jinghe population, but it had no significant effects on these factors in Yarkant. Similar inhibitory effects were found in the F_1_ and F_2_ generations, but the biological traits in the Yarkant population were significantly reduced when the parents (F_0_) were exposed to LC_25_ of acetamiprid, whereas the changes in the Jinghe population were not significant. However, sublethal sulfoxaflor showed a stimulatory effect on *A. gossypii* in the F_0_ and F_1_ generation; the adult fecundity and longevity of the F_0_ generation were significantly higher in Jinghe, while the biological traits of the F_1_ generation were obviously higher in Yarkant. In the F_2_ generation, the *r* and *λ* were significantly higher in Jinghe; meanwhile, these biological traits were reduced in Yarkant. These results indicate that sulfoxaflor and acetamiprid had different sublethal effects on *A. gossypii* that varied by generation. In addition, we speculate that the genetic background and the resistance levels of *A. gossypii* may also influence the sublethal effects. Our findings are useful for assessing the overall effects of sulfoxaflor and acetamiprid on *A. gossypii.*

## 1. Introduction

Physiological or behavioral alterations in individuals that survive exposure to insecticides at sublethal or lethal concentrations are characterized as sublethal effects [1]. Insects are frequently exposed to low or sublethal concentrations of insecticides in agro-ecosystems as a result of inappropriate insecticide application, drift, and insecticide degradation in the environment [1,2]. Sublethal effects are multispectral for insects, including impacts on development, sexual ratio, fecundity, longevity, feeding behavior, oviposition behavior, chemical communication, etc. [1,3,4,5,6]. Moreover, their offspring can be indirectly affected through transgenerational inheritance [7], which also potentially induces changes in communities and ecosystem services [8]. Therefore, sublethal effects are used as factors in assessing the environmental risk of insecticides to arthropods [9,10], and it is important to clarify the sublethal effects of insecticides on insects to facilitate the comprehensive evaluation and the rational use of insecticides.

Sublethal effects may be bidirectional (negative/positive) for pests exposed to sublethal/low concentrations of insecticides. Sublethal effects are generally detrimental to the ontogeny and reproduction of pests, for example, *Myzus persicae* exposed to LC_10_ and LC_25_ concentrations of sulfoxaflor [11], *Bradysia odoriphaga* exposed to sublethal (LC_5_ and LC_20_) concentrations of thiamethoxam [12], and *Bemisia tabaci* exposed to sublethal or low concentrations of four insecticides [13]. However, a positive effect on fertility has been observed in many pests, for instance, in *Aphis glycines* exposed to sublethal (LC_5_) concentration of imidacloprid [14] and in *M. persicae* exposed to low doses of imidacloprid [15]. This phenomenon is known as insecticide-induced hormesis, which is a stimulatory effect associated with low (sublethal) doses of an insecticide, and it is characterized by a reversal of the response between low and high doses of the insecticide [16,17]. Insecticide-induced hormesis has transgenerational effects, as was observed in the progeny generation (F_1_) of *Aphis gossypii*, when parental aphids (F_0_) were exposed to LC_15_ of thiamethoxam [18]. Moreover, the hormesis of the sublethal effect in resistant populations was demonstrated in pyrethroid-resistant weevils [19]. These results demonstrate that the sublethal effect of an insecticide can not only lead to chemical control failure but actually promote resistance evolution.

Many factors, such as different insecticides, pest species, and generations as well as the use of sublethal treatment concentrations, have an influence on sublethal effects [1]. For example, significantly higher fecundity was observed in *M. persicae* that were exposed to sublethal concentrations of acetamiprid and imidacloprid, but not in that exposed to sublethal concentrations of lambda cyhalothrin [20]. The sublethal concentrations of imidacloprid significantly reduced the adult longevity and fecundity of *A. gossypii* in the F_0_ generation, but they were significantly higher in the F_1_ generation [21]. In addition, variations in sublethal effects were observed in *A. gossypii*, *Nilaparvata lugens* and *Apolygus lucorum* after being treated with different sublethal concentrations of insecticides [22,23,24]. Additionally, in various populations of *M. persicae*, the F_0_ and F_1_ generations showed differing responses to LC_25_ concentrations of sulfoxaflor [25,26]. Similar results were found in different populations of *M. persicae* exposed to the same sublethal concentration of thiamethoxam [27,28]. Thus, the different genetic backgrounds of the pests, especially the different levels of resistance to insecticides, may also affect sublethal efficiency. Neonicotinoids, nicotinic acetylcholine receptor agonists, are widely used to control aphids, planthoppers, whiteflies, and other piercing-sucking pests [29]. Overreliance on these insecticides has resulted in the evolutionary resistance of *A. gossypii* to neonicotinoids [30,31]. Sulfoxaflor, a relatively new insecticide, has shown good control effects on numerous resistant pests due to the novelty of its chemical composition [32,33]. In China, sulfoxaflor has been a popular insecticide in recent years to control *A. gossypii*, *M. persicae*, *N. lugens*, and *B. tabaci* [34,35]. However, the field populations of *A. gossypii* and *N. lugens* in parts of China have developed a low level of resistance to sulfoxaflor since its application [36,37,38]. In addition, the sublethal effect of sulfoxaflor on *A gossypii*, *M. persicae*, and *N. lugens* has been reported in several studies [22,26,39]. These results indicate that these insects have a high risk of developing resistance to sulfoxaflor.

Recently, the proportion of cotton planting area in Xinjiang has increased year by year in China. The cotton aphid, *A. gossypii*, is a significant insect pest in Xinjiang, and it has developed resistance to a large number of frequently used insecticides. In this study, we compared the sublethal effects of acetamiprid and sulfoxaflor on *A. gossypii*, which was collected from two field populations with different genetic backgrounds in Xinjiang. An age-stage, two-sex life table was used to evaluate the sublethal effects of sulfoxaflor and acetamiprid on the life table parameters of *A. gossypii*, including development, survival, longevity, and fecundity. These findings will help to provide a comprehensive assessment of acetamiprid and sulfoxaflor for their use in integrated pest management and to provide a reference for their reasonable application in the field.

## 2. Materials and Methods

### 2.1. Insects

Two field populations of *A. gossypii* were collected in July 2019 and used in this study—one from Yarkant County (77°28′ E, 38°546′ N) of Xinjiang and the other from Jinghe County (82°896′ E, 44°588′ N) of Xinjiang. Then, they were taken back and reared on cotton seedlings, *Gossypium hirsutum*, in a greenhouse (26 ± 1 °C, 70 ± 5% RH, and 16:8 L/D) for further tests.

### 2.2. Insecticide and Reagents

Sulfoxaflor (95%) was provided by Corteva Agriscience (Indianapolis, IN, USA), and acetamiprid (97%) was provided by Jiangsu Weier Chemical Co., Ltd. (Yancheng, China). Triton X-100 was obtained from Beijing Coolaber Technology Co., Ltd. (Beijing, China). All chemicals and solvents were analytical grade reagents obtained from Sinopharm Chemical Reagent Co., Ltd. (Shanghai, China).

### 2.3. Toxicity Bioassays

Measurements of the toxicity of sulfoxaflor and acetamiprid to *A. gossypii* from Yarkant and Jinghe were carried out using the leaf dipping method, with slight modifications [40,41]. Sulfoxaflor and acetamiprid were dissolved separately in acetone to prepare stock solutions, and each was then diluted in a serially diluted using 0.05% (*v*/*v*) Triton X-100 in water to prepare five to six concentrations. The 23 mm diameter leaf discs, which were cut from fresh cotton leaves, were dipped for 15 s in the desired concentrations of insecticides or 0.05% (*v*/*v*) Triton X-100 (as a control), and then air-dried in the shade. The dried leaf discs were placed upside down in 12-well cell culture plates containing a 1% agar medium. A total of 30 apterous adult aphids were carefully transferred onto the leaf discs and then covered with Chinese art paper. Three independent biological replications were set for each concentration, and the mortality was assessed after 48 h. Bioassays were performed at 26 ± 1 °C, 70 ± 5% RH and at the 16:8 (L/D) photoperiod.

### 2.4. Sublethal Effects of Acetamiprid and Sulfoxaflor on A. gossypii

Three treatments, namely the control, sulfoxaflor and acetamiprid groups, were each set up in Yarkant and Jinghe. LC_25_ concentrations of sulfoxaflor and acetamiprid (obtained from the above experiments) were used to assess their sublethal effects on *A. gossypii* in different regions. All aphids were kept in a chamber at 26 ± 1 °C, 70 ± 5% RH with a 16:8 h (L/D) photoperiod. *A. gossypii* in each region was treated as follows.

F_0_ generation: Approximately 1000 apterous aphid adults were transferred to fresh cotton seedlings without any insecticides. After 24 h, all adults were removed, and the cotton seedlings containing nymphs were cultivated for 8 days to obtain adults at the same growth stage. About 200 apterous adults were treated with an LC_25_ concentration of sulfoxaflor or acetamiprid via the leaf dipping method, as described in Section 2.3, and 0.05% (*v*/*v*) Triton X-100 was used as a control. After 48 h, 100 survivors were selected (as the F_0_ generation) per group and transferred to insecticide-free leaf discs (23 mm) for individual rearing, and the leaves were placed upside down in 12-well cell culture plates containing 1% agar medium and then covered with nylon net. New leaf discs were renewed every 4 days until the adults died. The longevity and the fecundity of each group were recorded daily.

F_1_ and F_2_ generations: A total of 100 neonate nymphs per group were randomly selected from the former generation, and they were reared individually on insecticide-free leaf discs (as described in the F_0_ generation above). Life table parameters, including the longevity, fecundity, and development times of the different stages, were recorded daily until the death of the adults.

### 2.5. Statistical Analysis

A probit analysis was conducted with SPSS 25.0 (SPSS Inc., Chicago, IL, USA) to calculate the values of LC_25_ and LC_50_ (95% confidence intervals) concentrations of sulfoxaflor and acetamiprid. The longevity and the fecundity of the F_0_ generation were analyzed using Student’s t-tests, implementing SPSS 25.0. The raw data for each *A. gossypii* individual in the F_1_ and the F_2_ generations were analyzed via an age-stage two-sex life-table theory [42]; and, they were calculated using the TWOSEX-MS Chart program (Version 2021.10.30), including the pre-adult period, adult period, adult pre-oviposition period (APOP), total preoviposition period (TPOP), oviposition days, fecundity, intrinsic rate of increase (*r*), finite rate of increase (*λ*), net reproductive rate (*R_0_*), the mean generation time (*T*), gross reproduction rate (*GRR*), age-stage specific survival rates (*S_xj_*), age-specific survival rate (*l_x_*), age-specific fecundity (*m_x_*), age-specific net maternity (*l_x_m_x_*), and age-stage reproductive value (*v_xj_*). The mean and standard errors (SE) of the life table parameters were estimated by the bootstrap procedure in a TWOSEX-MS Chart with 100,000 random re-samplings, and the differences in the life table parameters were compared using the paired bootstrap test based on the confidence intervals of the differences implemented.

## 3. Results

### 3.1. Toxicity of Sulfoxaflor and Acetamiprid against A. gossypii

The probit analyses of sulfoxaflor and acetamiprid against *A. gossypii* adults from Yarkant and Jinghe after 48 h are summarized in Table 1. The LC_50_ values of acetamiprid were 5.66 and 35.93 mg·L^−1^ for Yarkant and Jinghe, respectively, and the resistance ratio of Jinghe was 6.35-fold compared to that of Yarkant. The LC_50_ values of sulfoxaflor against aphids of Yarkant and Jinghe were 3.42 and 11.16 mg·L^−1^, respectively, and the resistance ratio of Jinghe was 3.26-fold compared to that of Yarkant. The LC_25_ values of sulfoxaflor and acetamiprid for Yarkant and Jinghe aphids were chosen as the sublethal concentrations for the following experiments.

### 3.2. Sublethal Effects of Acetamiprid on A. Gossypii from Jinghe and Yarkant

#### 3.2.1. Sublethal Effects of Acetamiprid on the F_0_ Generation

The adult fecundity and longevity of the F_0_ generation treated with LC_25_ of acetamiprid in Jinghe and Yarkant are shown in Figure 1, respectively. In Jinghe, the adult longevity and fecundity of the F_0_ generation were significantly lower in the acetamiprid group than in the control group (fecundity: *t* = 2.386, *df* = 178, *p* = 0.018; longevity: *t* = 3.562, *df* = 198, *p* = 0.000). However, there were no significant differences in the adult fecundity and longevity of the F_0_ generation between the acetamiprid and the control groups in Yarkant (fecundity: *t* = 1.541, *df* = 186, *p* = 0.125; longevity: *t* = −1.329, *df* = 198, *p* = 0.186).

#### 3.2.2. Sublethal Effects of Acetamiprid on the F_1_ Generation

The duration of the development and the population parameters of the F_1_ generation in Jinghe and Yarkant are listed in Table 2. In Jinghe, the pre-adult period, adult period, adult pre-oviposition period (APOP), total preoviposition period (TPOP), oviposition days, total longevity, and fecundity were higher in the acetamiprid group compared to the control group, and there were no significant differences except in TPOP (Table 2 and Appendix A). The mean generation time (*T*) was significantly longer in the acetamiprid group than in the control group, and other population parameters, such as the intrinsic rate of increase (*r*), finite rate of increase (*λ*), net reproductive rate (*R_0_*), and gross reproduction rate (*GRR*), were not significantly different among the two treatments (Table 2 and Appendix A). The age-stage specific survival rate (*S_xj_* represents the probability that a newborn can develop to age x and stage j) and the age-specific survival rate (*l_x_*, the probability that a neonatal individual will survive to age x, ignoring the different stages) curves of the acetamiprid and the control groups were basically consistent, with no significant differences (Figure 2 and Figure 3, Jinghe). These results indicate that the LC_25_ concentration of acetamiprid did not significantly affect the aphid survival rate in the F_1_ generation. The data of age-specific fecundity (*m_x_*) and age-specific maternity (*l_x_m_x_*) showed that the highest fecundity peaks in the acetamiprid and the control groups occurred on Days 9 and 8, with 4.78 and 4.51 offspring, respectively (Figure 3, Jinghe). In addition, compared to the control group, the acetamiprid group had high values from Days 9 to 18, indicating that the LC_25_ concentration of acetamiprid may have a positive effect on fecundity in the F_1_ generation. The age-stage reproductive value (*v_xj_* represents the contribution of individuals of age x and stage j to the subsequent generation) in the acetamiprid group was markedly higher compared to the control group (Figure 4, Jinghe). The maximal *v_xj_* values were 13.38 and 12.24 in the acetamiprid and the control groups, respectively. These results indicate that the LC_25_ concentration of acetamiprid may have a positive effect on the subsequent generation.

In Yarkant, the adult period, total longevity, and fecundity were significantly reduced in the acetamiprid group compared to the control group, whereas no significant effects on the pre-adult period, APOP, TPOP, and oviposition days were observed in the F_1_ individuals (Table 2 and Appendix A). The *r*, *λ*, and *R_0_* of the acetamiprid group were significantly lower than those of the control group, but the *T* and *GRR* were not significantly different between the acetamiprid and the control groups (Table 2 and Appendix A). The adult *S_xj_* curves of the acetamiprid group decreased faster than those of the control group (Figure 2, Yarkant). Compared to the control group at Day 11, the *l_x_* curve of the acetamiprid group declined more rapidly (Figure 3, Yarkant). These results suggest that the LC_25_ concentration of acetamiprid may have negatively influenced the survival rate at the adult stage. The *m_x_* and *l_x_m_x_* markedly declined in the acetamiprid group (Figure 3, Yarkant). These results indicate that the LC_25_ concentration of acetamiprid reduced the fecundity level relative to the control level in *A. gossypii*. Comparing the patterns of the *v_xj_* of the control group, the values of *v_xj_* were obviously lower in the acetamiprid group (Figure 4, Yarkant). The maximal *v_xj_* values were 14.51 and 12.76 in the control and acetamiprid groups, respectively. These results indicate that the LC_25_ concentration of acetamiprid may have a negative effect on the subsequent generation.

#### 3.2.3. Sublethal Effects of Acetamiprid on the F_2_ Generation

In Jinghe, consistent with the observations for the F_1_ generation, there were no significant differences in the duration of the development, fecundity, and the population parameters between the acetamiprid and the control groups in the F_2_ generation (Table 2 and Appendix A). The patterns of the *S_xj_* curves of the two groups showed slight differences, with a higher survival rate for the nymph stages in the acetamiprid group (Figure 5, Jinghe). The *l_x_* curves from Day 9 onward were lower in the acetamiprid group than in the control group (Figure 6, Jinghe). These results indicate that the LC_25_ concentration of acetamiprid may have had a negative effect on the adult survival rate in the F_2_ generation. The patterns of the *m_x_* and *l_x_m_x_* curves of the acetamiprid and the control groups were similar (Figure 6, Jinghe). The *v_xj_* curves of the F_2_ generation were basically the same in the two treatments (Figure 7, Jinghe).

In Yarkant, the adult period, oviposition days, total longevity, and fecundity of the F_2_ generation were significantly reduced in the acetamiprid group compared to the control group, while significant differences in the pre-adult period, APOP, and TPOP were not observed (Table 2 and Appendix A). Apart from the *T*, the *r*, *λ*, and *GRR* were significantly decreased in the the acetamiprid group compared with the control group (Table 3 and Appendix A). The adult *S_xj_* curve decreased more rapidly in the acetamiprid group than in the control group (Figure 5, Yarkant). The *l_x_*, *m_x_*, and *l_x_m_x_* curves of the acetamiprid group were still lower than those of the control group for the F_2_ generation (Figure 6, Yarkant). These results indicate that the acetamiprid (LC_25_) still had a negative effect on the survival rate and the fecundity of the F_2_ generation. The *v_xj_* curves of the acetamiprid group were still lower than those of the control group for the F_2_ generation (Figure 7, Yarkant).

### 3.3. Sublethal Effects of Sulfoxaflor on A. Gossypii from Jinghe and Yarkant

#### 3.3.1. Sublethal Effects of Sulfoxaflor on the F_0_ Generation

The adult fecundity and the longevity of the F_0_ generation were significantly higher in the group treated with LC_25_ of sulfoxaflor in Jinghe compared with the control group (fecundity: *t* = −2.655, *df* = 181, *p* = 0.009; longevity: *t* = −2.488, *df* = 198, *p* = 0.014) (Figure 1). In contrast, no significant differences in the adult fecundity and longevity of the F_0_ generation were found in Yarkant (fecundity: *t* = −0.422, *df* = 174, *p* = 0.674; longevity: *t* = 0.343, *df* = 198, *p* = 0.732) (Figure 1).

#### 3.3.2. Sublethal Effects of Sulfoxaflor on the F_1_ Generation

No significant differences in the duration of the development and the fecundity of the F_1_ generation were observed between the sulfoxaflor and the control groups in Jinghe, but these were higher in the sulfoxaflor group (Table 3 and Appendix A). The *r*, *λ*, *R_0_*, *T*, and *GRR* were not significantly different between the sulfoxaflor and the control groups (Table 3 and Appendix A). The *S_xj_* and *l_x_* curves of the two treatments were basically consistent (Figure 2 and Figure 3, Jinghe). These results indicate that the LC_25_ concentration of sulfoxaflor did not significantly affect the aphid survival rate in the F_1_ generation. The *m_x_* and *l_x_m_x_* showed that the highest fecundity peaks in the sulfoxaflor group occurred on Day 7, with 4.89 offspring (Figure 3, Jinghe). In addition, compared to the control group, the sulfoxaflor group had high values of *m_x_* and *l_x_m_x_* from Day 7 to 12, indicating that the LC_25_ concentration of sulfoxaflor may have had a positive effect on fecundity in the F_1_ generation. The *v_xj_* in the sulfoxaflor group was markedly higher than in the control group (Figure 4, Jinghe), indicating that the LC_25_ concentration of sulfoxaflor may have a positive effect on the subsequent generation.

In addition to the pre-adult period, the APOP, and TPOP, adult period, total longevity, oviposition days, and fecundity of the F_1_ generation in Yarkant were significantly higher in the sulfoxaflor group compared to the control group (Table 3 and Appendix A). The *T* was significantly longer in the sulfoxaflor group than in the control group, and no significant differences in the *r*, *λ*, *R*_0_, or *GRR* were observed between the sulfoxaflor and the control groups. The adult *S_x_* curves of the sulfoxaflor and the control groups were similar in the first 13 days, but the curves in the sulfoxaflor group after 14 days were higher than those in the control group (Figure 2, Yarkant). Compared to the control group at Day 11, the *l_x_* curve of the sulfoxaflor group was higher (Figure 3, Yarkant). These results suggest that the LC_25_ concentration of sulfoxaflor may have had a positive effect. The *m_x_* and *l_x_m_x_* markedly rose in the sulfoxaflor group compared to the control group (Figure 3, Yarkant). These results indicate that the LC_25_ concentration of sulfoxaflor increased the fecundity level of *A. gossypii*. The values of *v_xj_* were markedly higher in the sulfoxaflor group, with a maximal *v_xj_* value of 16.73, compared to the patterns of the *v_xj_* in the control group (Figure 4, Yarkant). This result indicates that the LC_25_ concentration of sulfoxaflor may have had a positive effect on the subsequent generation.

#### 3.3.3. Sublethal Effects of Sulfoxaflor on the F_2_ Generation

Consistent with the F_1_ generation, there were no significant differences in the duration of the development and the fecundity of the F_2_ generation between the sulfoxaflor and the control groups in Jinghe (Table 3 and Appendix A). Compared to the control group, the *r* and *λ* were significantly higher and the *T* was significantly lower in the sulfoxaflor group. Other significant differences were not found in the two treatments (Table 3 and Appendix A). The patterns of *S_xj_* curves of the two groups showed slight differences, with the higher values of adult *S_xj_* in the first 8 days in the sulfoxaflor group (Figure 5, Jinghe). The *l_x_* curves for the sulfoxaflor group from Day 9 onward were lower than those for the control group (Figure 6, Jinghe). The patterns of *m_x_* and *l_x_m_x_* curves of the sulfoxaflor and the control groups showed slight differences, with the peaks for fecundity and high values from Day 5 to 7 in the sulfoxaflor group (Figure 6, Jinghe). The *v_xj_* curves of the F_2_ generation were basically the same in two groups (Figure 7, Jinghe).

In Yarkant, the measured values for the pre-adult period were significantly higher, and the fecundity was significantly lower in the F_2_ generation of the sulfoxaflor group compared to the control group (Table 3 and Appendix A). The *r* and *λ* of the sulfoxaflor group in the F_2_ generation were significantly lower compared to the control group (Table 3 and Appendix A). There were no significant differences in the other parameters. The *S_xj_*, *l_x_*, *m_x_*, *l_x_m_x_*, and *v_xj_* curves of the F_2_ generation of the sulfoxaflor group were not much different from those of the control group (Figure 5, Figure 6 and Figure 7, Yarkant). These results indicate that sulfoxaflor (LC_25_) had little or no effect on the F_2_ generation.

## 4. Discussion

Chemical control has been the main control measure for *A. gossypii* for many years. Acetamiprid is a common insecticide for the control of *A. gossypii*, but the resistance of *A. gossypii* to acetamiprid has emerged since its debut [30,43]. Compared to the recent monitoring results of *A. gossypii* resistance in Xinjiang [31], the resistance of cotton aphids to acetamiprid in both Yarkant and Jinghe in this study was low, but Jinghe aphids had high resistance to acetamiprid compared to Yarkant, i.e., the LC_50_ of acetamiprid in Jinghe aphids was 6.35 times that in Yarkant aphids. Compared to acetamiprid, *A. gossypii* in both Yarkant and Jinghe were more sensitive to sulfoxaflor, but the LC_50_ of sulfoxaflor in Jinghe aphids was 3.26 times that in Yarkant aphids. Thus, *A. gossypii* aphids collected from Yarkant were more sensitive to acetamiprid and sulfoxaflor than those collected from Jinghe. Because the sublethal effects of a low concentration of insecticides is known to vary among individuals depending on their susceptibility to the insecticide and the applied dose/concentration [17], we had selected these two field populations to investigate whether different genetic backgrounds and different insecticide-resistant *A. gossypii* affect the sublethal effect of insecticides.

Acetamiprid, a neonicotinoid, is a type of nicotinic acetylcholine receptor (nAChRs) agonist. Sulfoxaflor also acts on insect nAChRs [33,44], but the interaction of sulfoxaflor with insect nAChRs is distinct from that of neonicotinoids and other nAChR-acting insecticides, such as nicotine, spinosyns, nereistoxin analogs, and butenolides [33,45]. There have already been some reports about the sublethal effects of acetamiprid and sulfoxaflor on *A. gossypii* and *M. persicae*, but the results from different reports are quite different. For example, Ullah et al. [46] found an LC_15_ concentration of acetamiprid significantly reduced the longevity and fecundity of the parent generation (F_0_) *A. gossypii*. Shang et al. [47] found that the longevity of the parent generation (F_0_) of *A. gossypii* was significantly prolonged and the fecundity was higher in response to an LC_20_ concentration of sulfoxaflor. Sublethal concentrations (LC_10_ and LC_25_) of sulfoxaflor significantly inhibited the longevity and the fertility of *M.persicae* (F_0_) [26]. However, in other reports, no such sublethal effect was observed in terms of the longevity and fecundity of *A. gossypii* (F_0_) when exposed to an LC_25_ concentration of sulfoxaflor [39] or an LC_5_ concentration of acetamiprid [46], and of *M. persicae* (F_0_) when exposed to an LC_25_ concentration of sulfoxaflor [25]. Because the sublethal effect is influenced by many factors, different results from different reports are inevitable. In this study, we used the same treatment concentrations, and we tested the same *A. gossypii* population. Although the results of the *A. gossypii* populations collected from the two fields were not consistent, the general trend was that the LC_25_ concentration of sublethal sulfoxaflor appeared to have a stimulatory effect on *A. gossypii,* whereas sublethal acetamiprid showed adverse effects. Thus, sulfoxaflor presents a high risk of resistance evolution even though it is a novel insecticide. This is supported by other findings in which the field populations of *A. gossypii* and *N*. *lugens* in parts of China were found to have developed a low level of resistance to sulfoxaflor since it was approved for use in 2014 [36,37,38]. In the future, we should use sulfoxaflor more rationally and sparingly to extend its service life.

To further reveal the effects of genetic background on sublethal efficiency, the difference in responses to being treated with sublethal concentrations of insecticides were compared between the two field *A. gossypii* populations. Significantly adverse effects of acetamiprid on the longevity and the fecundity of the F_0_ generation of *A. gossypii* were observed in Jinghe, and hormesis was observed in F_0_ adults of Jinghe after exposure to the LC_25_ concentration of sulfoxaflor, whose longevity and fecundity were significantly stimulated. However, no significant sublethal effects of acetamiprid and sulfoxaflor on F_0_ adults were found in Yarkant. Previously, Guedes et al. [17] analyzed the reason differing sublethal effects of the same insecticide. They believe the main factor is the response of insect individuals to the pressure from the insecticide (sublethal), which varies between lower and higher thresholds in a dose/concentration-dependent manner (i.e., the basic dose/concentration response relationship of toxicology), and individuals may be hardly or not at all affected by exposure. According to the results of our study, we expect that the sublethal effects of insecticide may not only be influenced by the different responses of insect individuals but also related to differences in genetic backgrounds, especially with regard to the sensitivity level of the tested population.

Sublethal effects across generations have been observed in a wide range of pests after exposure to sublethal concentrations of insecticides [11,21,22]. Aphids have the characteristic of telescoping generations, i.e., newborn individuals are born with embryos that also contain embryos [48], so the offspring can ingest the insecticides through the mother’s body [11,15]. This transgenerational effect on the F_1_ generation was observed in the Yarkant population, hormesis was observed in the F_1_ generation of the sulfoxaflor group, and negative sublethal effects occurred in the F_1_ generation of the acetamiprid group. Several other studies have shown that neonicotinoid insecticides have negative sublethal effects on many pests, such as *N. lugens* [22], *Sitobion avenae* [49], *A. glycines* [14], and *A. lucorum* [24,50]. However, it was found that an LC_15_ concentration of acetamiprid had positive sublethal effects on exposed *A. gossypii* [46]. Conversely, in this study, significant transgenerational effects on the F_1_ generations of the acetamiprid and sulfoxaflor groups were not observed in Jinghe even though the parents (F_0_) were subjected to significant sublethal effects of sulfoxaflor (stimulatory) and acetamiprid (negative), respectively. The lack of a transgenerational effect was also observed in *N. lugens* exposed to an LC_15_ concentration of sulfoxaflor [22] and *A. gossypii* exposed to LC_5_ concentrations of clothianidin and acetamiprid [21,23]. The offspring of treated F_0_ *A. gossypii* also showed a dose/concentration-dependent response, so the difference in offspring taking up amounts of insecticide from the mother was a key factor that influenced the transgenerational effect of the F_1_ generation. In addition, it is possible that differences in the resistance of *A. gossypii* contribute to differences in transgenerational effects. Detoxification enzymes could be induced after exposure to sublethal concentrations of insecticides, which may still occur in insecticide-resistant populations [17]. This induction may make it easy for insecticide-resistant populations to escape the effects of this insecticide pressure, as in the case of the F_1_ generations in Jinghe, which had relatively high resistance to sulfoxaflor and acetamiprid.

The transgenerational effect of acetamiprid was the same for the F_1_ and the F_2_ generations in Yarkant and Jinghe. Just as with the F_1_ generation, a negative transgenerational effect on the F_2_ generation was still observed in the acetamiprid group of Yarkant, and a significant transgenerational effect on the F_2_ generation of the acetamiprid was not observed in Jinghe. However, the transgenerational effect of sulfoxaflor varied across generations in Yarkant and in Jinghe. The transgenerational effect of sulfoxaflor on the F_2_ generation was different from that on the F_1_ generation in the sulfoxaflor group of Yarkant, e.g., there was a significant decrease in the fecundity, *r*, and *λ* of the F_2_ generation. Conversely, hormesis of sulfoxaflor was observed in the F_2_ generation of the sulfoxaflor group of Jinghe, e.g., there was a significant increase in the *r* and *λ*. These changes may be related to physiological trade-offs [16], and differences in the expression of these trade-offs may vary within and between generations [15].

It is known that the changes of biological parameters in insects is caused by genetic alteration and the fitness change with its resistance level to insecticides. For example, Valmorbida et al. found that *A. glycines* carrying the heterozygous super-kdr M918I+L1014F genotype had significant reproductive advantages [51]. The relative fitness of pyrethroid-resistant and organophosphate-resistant *Sitophilus zeamais* varied in different geographical populations [52,53]. We found that there were differences in the sublethal effects of acetamiprid and sulfoxaflor on Jinghe and Yarkant *A. gossypii* (F_0_, F_1_ and F_2_) in this paper. However, which genetic changes in the two tested populations are related with their suscepbility to insecticides and resulted in their different responses to sublethal concentration insecticides is unclear, and it needs to be further studied in the future.

Chemical control is an important control tool in integrated pest management (IPM). Due to the excessive reliance on chemical control, cotton pests such as *A. gossypii*, *B. tabaci*, and *A. lucorum* have serious resistance problems [54]. Pest resistance problems often lead to IPM failure. Monitoring the resistance of cotton pests to insecticides, and using various resistance detection techniques to grasp the changes in the resistance level can provide a basis for developing a resistance management strategy and the precision application of insecticides against cotton pests. Our results showed the sublethal effects of sulfoxaflor and acetamiprid varied according to differences in *A. gossypii* populations and generations. The genetic backgrounds and the resistance levels of *A. gossypii* may influence the sublethal effects of insecticides. These results suggest that field resistance monitoring should be carried out not only for different insecticides but also for different sites with varies basic resistance levels. In addition to chemical control, cotton pests can be controlled by ecological regulation methods. For example, safflowers (*Carthamus tinctorius*) are used to trap *Lygus pratensis* in cotton fields [55]. Rape (*Brassica napus*) and alfalfa (*Medicago sativa*) are used to attract natural enemies to control *A. gossypii* [56]. Therefore, environmentally friendly control strategies such as agricultural control, physical control, and biological control should be used to control cotton pests, reduce dependence on insecticides, and delay the development of the resistance of cotton pests to insecticides.

## 5. Conclusions

The sublethal effects of sulfoxaflor and acetamiprid vary according to differences in *A. gossypii* populations and generations. The genetic backgrounds and the resistance levels of *A. gossypii* may influence the sublethal effects of insecticides. In Jinghe, exposure of *A. gossypii* parents (F_0_) to sulfoxaflor and acetamiprid could be beneficial for newborn offspring (both F_1_ and F_2_ generations), whereby they are more likely to overcome the stress caused by sulfoxaflor and acetamiprid due to having a relatively high resistance. Moreover, hormesis was observed in the F_0_ and F_2_ generations of Jinghe and in the F_1_ generation of Yarkant when their parents were exposed to sulfoxaflor, which may lead to pest resurgence. Even though sulfoxaflor was very toxic against *A. gossypii* in the present study, insecticide-induced hormesis should be taken into consideration, particularly in insecticide-resistant *A. gossypii* populations. Our study focused on assessing the sublethal effects of sulfoxaflor and acetamiprid on differently insecticide-resistant *A. gossypii*, and these findings allow a more comprehensive understanding of the sublethal effects of insecticides.

## Figures and Tables

**Figure 1 insects-13-00498-f001:**
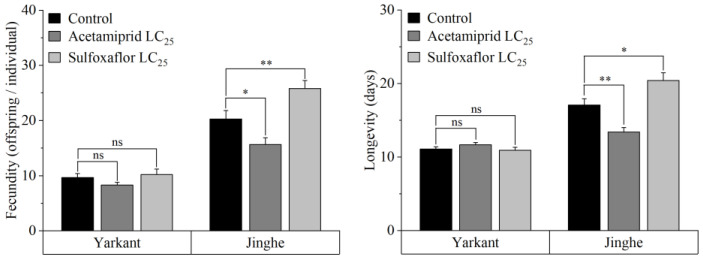
Effects of sublethal concentrations of acetamiprid and sulfoxaflor on the fecundity and longevity of the F_0_ generation in Yarkant and Jinghe. The data are represented as mean ± SE. Asterisks above the bars correspond to significant differences between the treatment and the control based on Student’s *t*-tests (*, *p* < 0.05; **, *p* < 0.01).

**Figure 2 insects-13-00498-f002:**
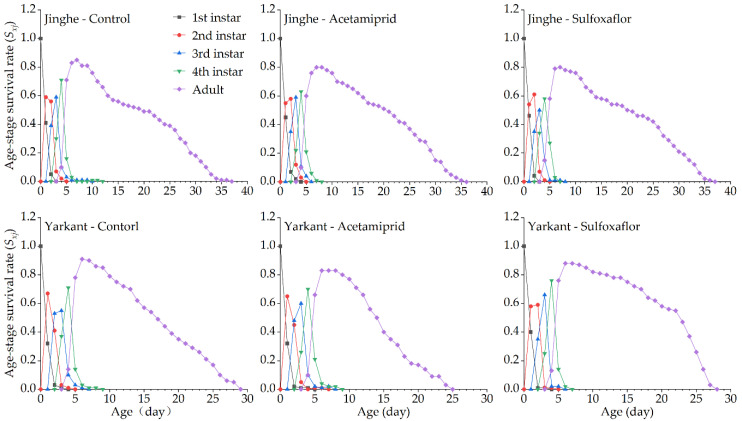
Effects of sublethal concentrations (LC_25_) of acetamiprid and sulfoxaflor on the age-stage specific survival rate (*S_xj_*) of *A. gossypii* of the F_1_ generation in Yarkant and Jinghe.

**Figure 3 insects-13-00498-f003:**
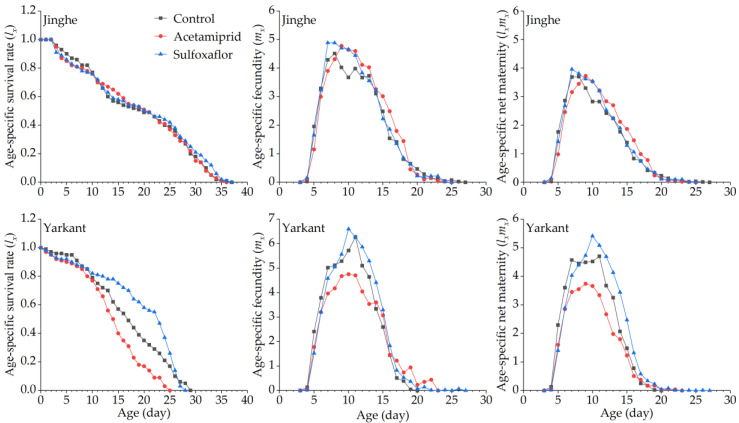
Effects of sublethal concentrations (LC_25_) of acetamiprid and sulfoxaflor on the age-specific survival rate (*l_x_*), age-specific fecundity (*m_x_*), and age-specific maternity (*l_x_m_x_*) of *A. gossypii* of the F_1_ generation in Yarkant and Jinghe.

**Figure 4 insects-13-00498-f004:**
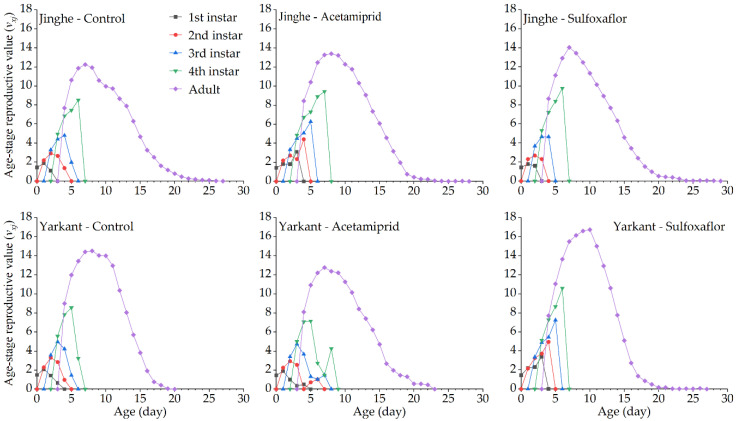
Effects of sublethal concentrations (LC_25_) of acetamiprid and sulfoxaflor on the age-stage reproductive value (*v_xj_*) of *A. gossypii* of the F_1_ generation in Yarkant and Jinghe.

**Figure 5 insects-13-00498-f005:**
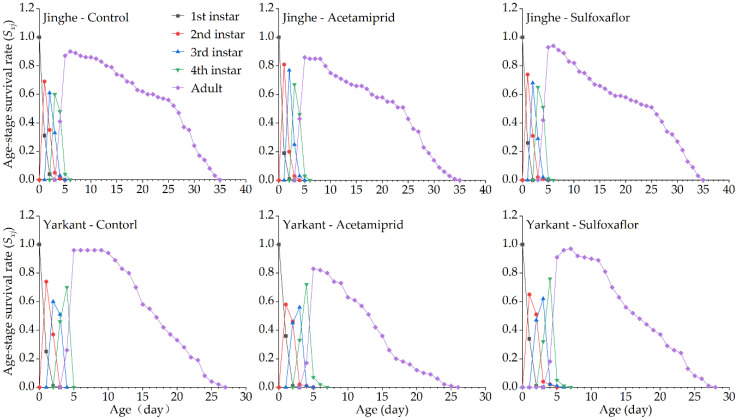
Effects of sublethal concentrations (LC_25_) of acetamiprid and sulfoxaflor on the age-stage specific survival rate (*S_xj_*) of *A. gossypii* of the F_2_ generation in Yarkant and Jinghe.

**Figure 6 insects-13-00498-f006:**
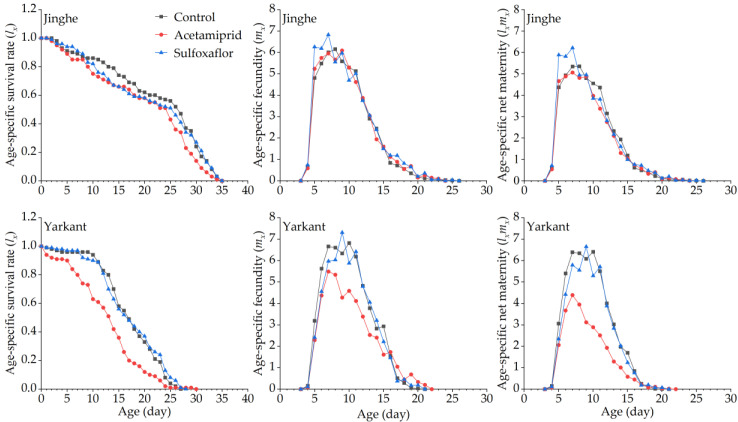
Effects of sublethal concentrations (LC_25_) of acetamiprid and sulfoxaflor on the age-specific survival rate (*l_x_*), age-specific fecundity (*m_x_*), and age-specific maternity (*l_x_m_x_*) of *A. gossypii* of the F_2_ generation in Yarkant and Jinghe.

**Figure 7 insects-13-00498-f007:**
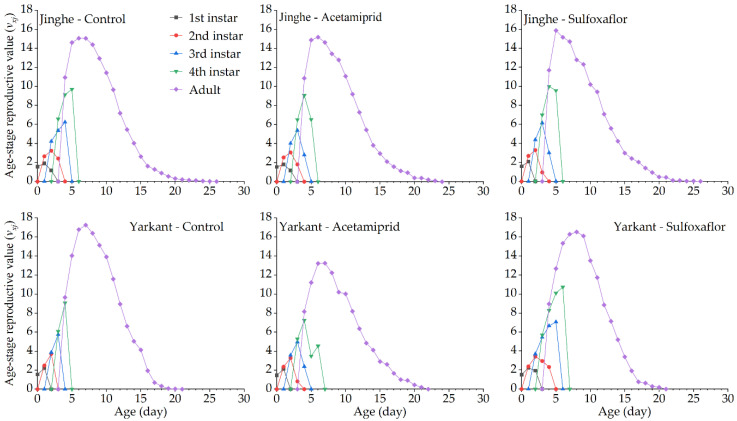
Effects of sublethal concentrations (LC_25_) of acetamiprid and sulfoxaflor on the age-stage reproductive value (*v_xj_*) of *A. gossypii* of the F_2_ generation in Yarkant and Jinghe.

**Table 1 insects-13-00498-t001:** Toxicity of sulfoxaflor and acetamiprid to *Aphis gossypii* in Yarkant and Jinghe.

Insecticide	Region	Slope ± SE ^a^	LC_25_ mg·L^−1^ (95%CI) ^b^	LC_50_ mg·L^−^^1^ (95%CI)	*RR* ^c^	*χ*^2^ (*df*)	*p*
Sulfoxaflor	Yarkant	1.51 ± 0.12	1.22 (0.86–1.63)	3.42 (2.65–4.35)	1	14.35 (13)	0.35
	Jinghe	1.02 ± 0.10	2.44 (1.47–3.61)	11.16 (8.03–15.38)	3.26	9.55 (13)	0.73
Acetamiprid	Yarkant	2.24 ± 0.18	2.83 (2.19–3.50)	5.66 (4.63–6.90)	1	15.48 (13)	0.28
	Jinghe	1.50 ± 0.14	12.77 (8.71–16.99)	35.93 (28.65–43.81)	6.35	13.30 (16)	0.65

^a^ Standard error. ^b^ 95% confidence intervals. ^c^ *RR* (resistance ratio) = LC_50_ of Jinghe/LC_50_ of Yarkant.

**Table 2 insects-13-00498-t002:** Sublethal effects of acetamiprid on the duration of development and the population parameters of the F_1_ and F_2_ generations in Yarkant and Jinghe.

Stages	F_1_ Generation				F_2_ Generation			
	Jinghe		Yarkant		Jinghe		Yarkant	
	Control	Acetamiprid	Control	Acetamiprid	Control	Acetamiprid	Control	Acetamiprid
Pre-adult (days)	5.10 ± 0.07 a	5.23 ± 0.09 a	5.01 ± 0.06 a	5.15 ± 0.08 a	4.59 ± 0.06 a	4.53 ± 0.06 a	4.73 ± 0.05 a	4.85 ± 0.05 a
Adult (days)	16.51 ± 0.96 a	17.54 ± 0.90 a	13.33 ± 0.67 a	10.87 ± 0.50 b	20.11 ± 0.83 a	18.08 ± 0.88 a	13.31 ± 0.47 a	9.80 ± 0.55 b
APOP (days)	0.26 ± 0.05 a	0.40 ± 0.05 a	0.26 ± 0.05 a	0.26 ±0.05 a	0.22 ± 0.05 a	0.24 ± 0.05 a	0.34 ± 0.05 a	0.35 ± 0.05 a
TPOP (days)	5.36 ± 0.07 b	5.62 ± 0.09 a	5.29 ± 0.08 a	5.41 ± 0.08 a	4.81 ± 0.06 a	4.77 ± 0.06 a	5.07 ± 0.04 a	5.18 ± 0.04 a
Oviposition days	9.71 ± 0.47 a	10.94 ± 0.42 a	9.37 ± 0.37 a	8.85 ± 0.42 a	11.20 ± 0.34 a	10.77 ± 0.46 a	10.29 ± 0.23 a	7.92 ± 0.41 b
Total longevity (days)	21.61 ± 0.95 a	22.77 ± 0.89 a	18.34 ± 0.66 a	16.02 ± 0.49 b	24.70 ± 0.83 a	22.61 ± 0.87 a	18.04 ± 0.47 a	14.65 ± 0.55 b
Fecundity (offspring/individual)	36.86 ± 2.05 a	41.93 ± 1.71 a	44.44 ± 2.42 a	36.68 ± 2.21 b	49.35 ± 1.56 a	47.45 ± 2.07 a	53.42 ± 1.29 a	33.40 ± 1.90 b
Parameters								
*r* (d^−1^)	0.363 ± 0.008 a	0.350 ± 0.008 a	0.396 ± 0.007 a	0.363 ± 0.008 b	0.441 ± 0.006 a	0.438 ± 0.007 a	0.436 ± 0.004 a	0.378 ± 0.008 b
*λ* (d^−1^)	1.438 ± 0.011 a	1.419 ± 0.011 a	1.486 ± 0.011 a	1.437 ± 0.012 b	1.554 ± 0.010 a	1.549 ± 0.016 a	1.547 ± 0.007a	1.459 ± 0.001 b
*R_0_* (offspring/individual)	31.700 ± 2.180 a	33.960 ± 2.153 a	40.434 ± 2.532 a	31.278 ± 2.299 b	44.912 ± 2.011 a	41.756 ± 2.378 a	51.280 ± 1.623 a	28.388 ± 2.005 b
*T* (days)	9.522 ± 0.122 b	10.066 ± 0.130 a	9.337 ± 0.095 a	9.493 ± 0.105 a	8.633 ± 0.081 a	8.526 ± 0.075 a	9.025 ± 0.073 a	8.850 ± 0.098 a
*GRR* (offspring/individual)	44.336 ± 1.720 a	47.789 ± 1.261 a	51.810 ± 1.999 a	46.969 ± 1.954 a	53.795 ± 1.225 a	53.526 ± 1.457 a	58.341 ± 1.099 a	44.925 ± 1.970 b

Mean ± SE were estimated using the bootstrap technique with 100,000 re-samplings. Different letters in rows indicate significant differences between the acetamiprid and the control groups at *p* < 0.05, paired bootstrap test using the TWOSEX-MS Chart program.

**Table 3 insects-13-00498-t003:** Sublethal effects of sulfoxaflor on the duration of development and the population parameters of F_1_ and F_2_ generation in Yarkant and Jinghe.

Stages	F_1_ Generation				F_2_ Generation			
	Jinghe		Yarkant		Jinghe		Yarkant	
	Control	Sulfoxaflor	Control	Sulfoxaflor	Control	Sulfoxaflor	Control	Sulfoxaflor
Pre-adult (days)	5.10 ± 0.07 a	5.17 ± 0.08 a	5.01 ± 0.06 a	5.04 ± 0.06 a	4.59 ± 0.06 a	4.56 ± 0.05 a	4.73 ± 0.05 b	4.89 ± 0.05 a
Adult (days)	16.51 ± 0.96 a	17.57 ± 1.01 a	13.33 ± 0.67 b	16.21 ± 0.63 a	20.11 ± 0.83 a	18.33 ± 0.96 a	13.31 ± 0.47 a	13.01 ± 0.56 a
APOP (days)	0.26 ± 0.05 a	0.32 ± 0.06 a	0.26 ± 0.05 a	0.33 ± 0.06 a	0.22 ± 0.05 a	0.21 ± 0.04 a	0.34 ± 0.05 a	0.26 ± 0.05 a
TPOP (days)	5.36 ± 0.07 a	5.48 ± 0.09 a	5.29 ± 0.08 a	5.36 ± 0.06 a	4.81 ± 0.06 a	4.78 ± 0.05 a	5.07 ± 0.04 a	5.14 ± 0.05 a
Oviposition days	9.71 ± 0.47 a	10.56 ± 0.45 a	9.37 ± 0.37 b	11.19 ± 0.33 a	11.20 ± 0.34 a	10.64 ± 0.44 a	10.29 ± 0.23 a	9.61 ± 0.31 a
Total longevity (days)	21.61 ± 0.95 a	22.74 ± 1.00 a	18.34 ± 0.66 b	21.25 ± 0.62 a	24.70 ± 0.83 a	22.89 ± 0.94 a	18.04 ± 0.47 a	17.90 ± 0.56 a
Fecundity (offspring/individual)	36.86 ± 2.05 a	41.19 ± 1.84 a	44.44 ± 2.42 b	51.48 ± 1.99 a	49.35 ± 1.56 a	49.45 ± 2.12 a	53.42 ± 1.29 a	48.58 ± 1.84 b
Parameters								
*r* (d^−1^)	0.363 ± 0.008 a	0.364 ± 0.009 a	0.396 ± 0.007 a	0.380 ± 0.006 a	0.441 ± 0.006 b	0.462 ± 0.008 a	0.436 ± 0.004 a	0.418 ± 0.005 b
*λ* (d^−1^)	1.438 ± 0.011 a	1.440 ± 0.013 a	1.486 ± 0.011 a	1.462 ± 0.009 a	1.554 ± 0.010 b	1.587 ± 0.013 a	1.547 ± 0.007a	1.519 ± 0.007 b
*R_0_* (offspring/individual)	31.700 ± 2.180 a	33.630 ± 2.202 a	40.434 ± 2.532 a	45.294 ± 2.420 a	44.912 ± 2.011 a	46.481 ± 2.303 a	51.280 ± 1.623 a	47.120 ± 1.965 a
*T* (days)	9.522 ± 0.122 a	9.624 ± 0.147 a	9.337 ± 0.095 b	10.034 ± 0.073 a	8.633 ± 0.081 a	8.312 ± 0.092 b	9.025 ± 0.073 a	9.221 ± 0.070 a
*GRR* (offspring/individual)	44.336 ± 1.720 a	47.106 ± 1.330 a	51.810 ± 1.999 a	55.568 ± 1.380 a	53.795 ± 1.225 a	56.472 ± 1.707 a	58.341 ± 1.099 a	55.609 ± 1.346 a

Mean ± SE were estimated using the bootstrap technique with 100,000 re-samplings. Different letters in rows indicate significant differences between the sulfoxaflor and the control groups at *p* < 0.05, paired bootstrap test using the TWOSEX-MS Chart program.

## Data Availability

The data presented in this study are available on request from the corresponding authors.

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
