# Peer review of "Differences in the Sublethal Effects of Sulfoxaflor and Acetamiprid on the Aphis gossypii Glover (Homoptera: Aphididae) Are Related to Its Basic Sensitivity Level"

_insects, 2022, doi:10.3390/insects13060498_

Round 1

Reviewer 1 Report

The topic of the paper is really tricky and interesting. It is difficult to monitor resistance insurgence on the wild population, may be a monitoring net could be usefull, even if costly.

Author Response

Thank you very much for your careful consideration of our manuscript. In accordance with your comments, the grammar and spelling errors in the full text have been corrected.

Reviewer 2 Report

The paper ‘Differences in the sublethal effects of sulfoxaflor and acetamiprid on Aphis gossypii…’ investigates the transgenerational effects of insecticide application on two genetically distinct field populations of cotton aphid, measured by fecundity and mortality of adults. The authors identified a difference in susceptibility to the two compounds with sulfoxaflor showing a higher toxicity than acetamiprid, in addition to some differences found between the two populations. Analysis of the transgenerational effects of sublethal insecticide application was performed on the F1 and F2 generations of the populations. This identified no significant changes in the Yarkant population but saw some fluctuations in the fitness of the Jinghe population.

While the paper is well-written, well-structured and the science generally experimentally sound, there are some questions on the novelty of this study. I therefore recommend to reject this manuscript for the reasons below.

The study measures the fecundity and mortality of the two populations and identifies some small significant differences in the Yarkant population. Previous studies (Sublethal and transgenerational effects of sulfoxaflor on the biological traits of the cotton aphid Aphis gossyppi) and (Toxicities and sublethal effects of seven neonicotinoid insecticides on survivial, growth and reproduction of imidacloprid-resistant cotton aphid Aphis gossyppi), among others, have previously characterised some sublethal effects of both acetamiprid and sulfoxaflor in the same species presented here. Small differences are described between the studies but this may well be due to experimental variability. For this reason, I do not believe that this study demonstrates significant novelty to enable publication.

To improve the study, I would suggest attempting to explain why the effects seen here are occurring rather than simply describing the bioassay outcomes, especially considering similar studies have previously been published. For example, why is there a difference between the two populations of cotton aphid. How genetically different are they?

Similarly, the manuscript identifies these small changes in fitness of the population but makes no effort to identify why this is. The paper speculates that changes in gene expression could be responsible, but there is no attempt to characterise this.

Author Response

Dear Reviewer

Thank you very much for your attention and careful consideration to our manuscript.

Although previous studies studied the sublethal effects of insecticides (including sulfoxaflor and acetamiprid) on the biological traits of the cotton aphid, but the results from different reports were quite different. So, in this study, we focused on the differences of sublethal effects influenced by the genetic background and resistance levels of A. gossypii populations. The life table method was used to analyze the sublethal effects of insecticides on two different cotton aphid populations. While there may be errors in the trial, but we tried to compare the differences between two A. gossypii populations in same conditions, such as same experiment condition, same insecticide, and sublethal concentrations of insecticides. So, we think the differences are meaningful, which can partly explain the reasons for the differences in the results in previous reports. And from our results, we speculate that the genetic background and resistance levels of A. gossypii may also influence the sublethal effects, it will be useful for assessing the overall effects of sulfoxaflor and acetamiprid on A. gossypii.

Thank you for your suggestion that we should further analyze what causes differences in fitness between populations in terms of genetic differences. This is also our next step in our research work. In the discussion section, we have added some discussion on this issue.

Reviewer 3 Report

Differences in the sublethal effects of sulfoxaflor and acetamiprid on Aphis gossypii Glover (Homoptera: Aphididae) are related to the basic sensitivity level

General Comments:

The management of cotton aphid is timely topic that impacts the integrated pest management (IPM) practices of cotton crop globally. Management practices also known to start and remain, for some time, locally in certain geographical areas until being adopted after proved to be successful. It is important to follow clear approach to have better understanding of the whole situation and achieve solid conclusion. The methods and conclusions of this study are appropriate. The continuous monitoring of efficacy and resistant of selective chemistries, like sulfoxaflor, is important to ensure proper management of certain sucking mouth insects in cotton. The manuscript provided insight into this topic, and possibility of managing resistant on aphid in cotton.

The manuscript suffers from some “add on”, especially the results section.

Methods and number of the same parameter changed over time and locations. 

Some section in the M&M are not used to obtain results.

Some of the statistical analyses used are not the best fit to compare data, but could be acceptable.

The paper has some structural and grammar flaws. Data supporting the conclusion, but I think the manuscript could still benefit from focus on the specific method of monitoring in the geographical area of the study. Some discussion about other cotton pests, and approaches of controlling them could enhance the paper and introduce broader concept of the issue of IPM in cotton. 

Author Response

Dear Reviewer

Thank you very much for your attention and careful consideration to our manuscript. We have revised the manuscript according to your suggestions and comments. The main comments and our responses to them are given below:

Point 1: Some section in the M&M are not used to obtain results.

Response 1: The parameters of the evaluation have been redefined in Line 173-174.

Point 2: The paper has some structural and grammar flaws. Data supporting the conclusion, but I think the manuscript could still benefit from focus on the specific method of monitoring in the geographical area of the study. Some discussion about other cotton pests, and approaches of controlling them could enhance the paper and introduce broader concept of the issue of IPM in cotton.

Response 2: In accordance with your comments, the grammar and spelling errors in the full text have been corrected. In addition, the discussions of resistance monitoring and control in other cotton pests have added in the discussion section (Line 470-482).

Round 2

Reviewer 2 Report

In light of the response to the comments in addition to other reviewers, I recommend that this manuscript is published in it's current form.